# Thyroglossal Duct Lipoma: A Case Report and a Systematic Review of the Literature for Its Management

**DOI:** 10.3390/diagnostics13050932

**Published:** 2023-03-01

**Authors:** Luca Giovanni Locatello, Marilena Graziadio, Elena D’Orlando, Alfredo Vallone, Cesare Miani, Enrico Pegolo, Maria Gabriella Rugiu

**Affiliations:** 1Department of Otorhinolaryngology, Academic Hospital “Santa Maria della Misericordia”, Azienda Sanitaria Universitaria Friuli Centrale, Piazzale Santa Maria della Misericordia 15, 33100 Udine, Italy; 2Department of Otorhinolaryngology, Sant’Antonio Abate Hospital, Azienda Sanitaria Universitaria Friuli Centrale, 33028 Tolmezzo, Italy; 3Department of Medicine (DAME), University of Udine, Via Colugna 50, 33100 Udine, Italy; 4Institute of Anatomic Pathology, Academic Hospital “Santa Maria della Misericordia”, Azienda Sanitaria Universitaria Friuli Centrale, Piazzale Santa Maria della Misericordia 15, 33100 Udine, Italy

**Keywords:** neck disorders, lipoma, thyroglossal duct, dysphagia, Sistrunk operation

## Abstract

Thyroglossal duct (TGD) remnants in the form of cysts or fistulas usually present as midline neck masses and they are removed along with the central body of the hyoid bone (Sistrunk’s procedure). For other pathologies associated with the TGD tract, the latter operation might be not necessary. In the present report, a case of a TGD lipoma is presented and a systematic review of the pertinent literature was performed. We present the case of a 57-year-old woman with a pathologically confirmed TGD lipoma who underwent transcervical excision without resecting the hyoid bone. Recurrence was not observed after six months of follow-up. The literature search revealed only one other case of TGD lipoma and controversies are addressed. TGD lipoma is an exceedingly rare entity whose management might avoid hyoid bone excision.

**Figure 1 diagnostics-13-00932-f001:**
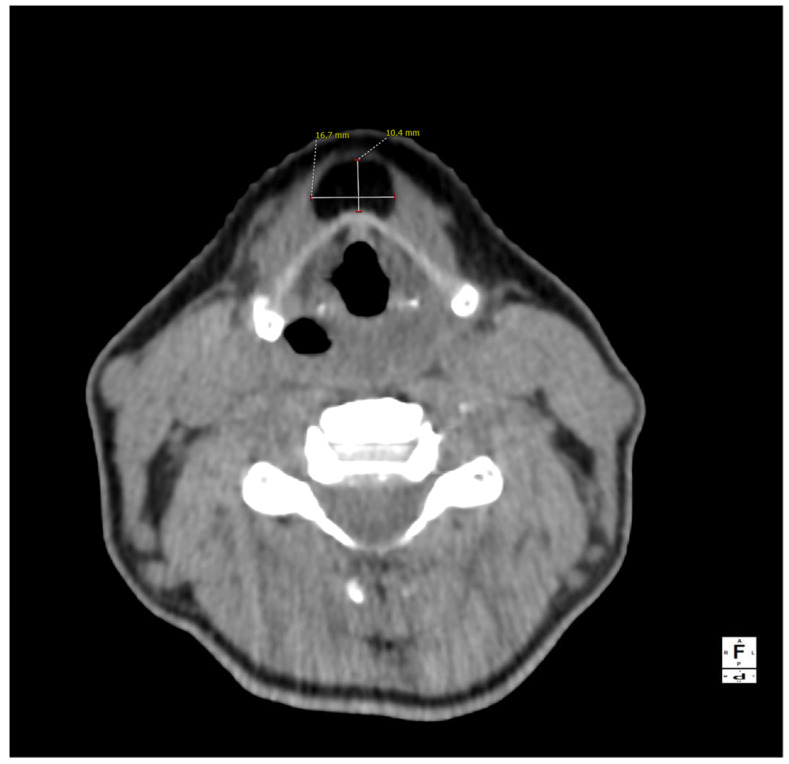
A computed tomography of the neck on the axial plane demonstrating a 10 × 17 × 38 mm (AP × LL × CC) lesion and its relationships with the anterior border of the thyroid cartilage of the larynx. The anatomical subsite pinpoints to a possible persistence of the thyroglossal duct (TGD). This represents an embryological remnant during the migration of the endodermal thyroid diverticulum, and it may extend from the foramen cecum of the base of the tongue to the suprasternal notch [1,2,3]. TGD can variably persist in up to 7% of the adult human population, and its incomplete obliteration (usually by the tenth week of gestation) leads to the creation of the pyramidal lobe of the thyroid gland as well as many other neck disorders, including lingual thyroiditis and TGD cyst [4]. The latter in particular is the most frequent congenital central neck mass and it usually presents around 30–40 years of age as an infected cyst (more rarely as a fistula or a sinus), causing neck discomfort and dysphagia [5,6]. Their treatment usually consists of open surgical excision along with the central portion of the hyoid bone’s body and a core of tongue musculature, as described in 1920 by Walter E. Sistrunk from the Mayo Clinic [7]. The differential diagnosis of TGD cysts and midline masses is vast and it includes, among others, dermoid and epidermoid cysts, cystic nodal metastases of papillary thyroid carcinoma, lymphangiomas, lymphomas, ectopic thyroid adenomas, or lipomas [4,8].

**Figure 2 diagnostics-13-00932-f002:**
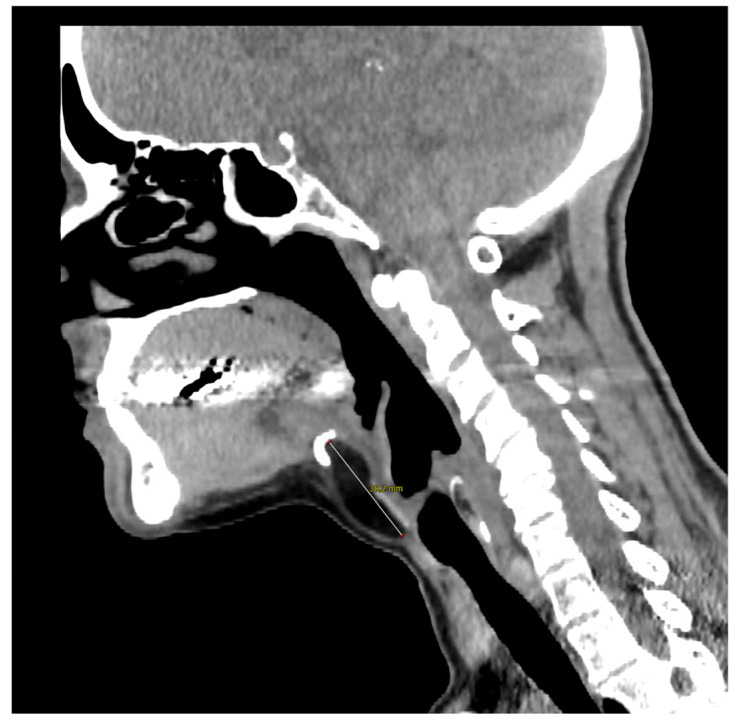
Computed tomography of the neck on the sagittal plane illustrates the close contact with the hyoid bone whose resection was not necessary in the present case. The 57-year-old woman came to our outpatient clinic because of a slow-growing median cervical swelling that had been present for more than 14 months. She complained of mild neck discomfort without frank dysphagia or dyspnea. Upon physical examination, the subcutaneous swelling was soft and nontender, it was partially mobile on swallowing, and there were no signs of infection or inflammation. Her medical history included seasonal allergic rhinitis, nodular osteoarthritis of the hands, primary Sjogren’s syndrome, and a recent diagnosis of euthyroid Hashimoto’s thyroiditis. There were no known allergies nor she was taking any medication. A full ear, nose, and throat evaluation was uneventful, and the cranial nerves function assessment was normal. In particular, transnasal fiberoptic flexible examination revealed regular oropharyngeal and laryngeal structures with no pooling secretions, and normal swallowing phases for both solids and liquids. In the suspicion of a TGD cyst, an initial ultrasound (US) examination was requested and it revealed a non-cystic hypoechoic mass with scattered hyperechoic spots and a normal-looking thyroid gland; no lymphadenopathies were present. Subsequently, CT and MRI were both ordered in the diagnostic work-up.

**Figure 3 diagnostics-13-00932-f003:**
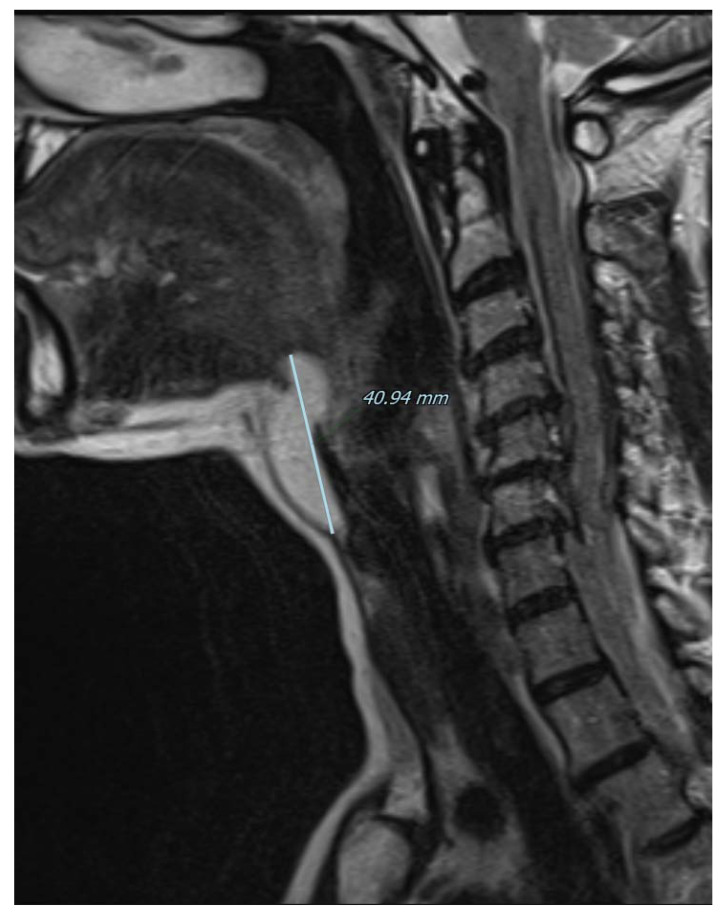
Magnetic resonance imaging of the neck on a coronal plane, T2-weighted, showing the craniocaudal extension of the mass from the base of the tongue to the anterior neck region. Picture was highly suggestive of lipoma, therefore fine needle aspiration was not performed and the patient was referred for surgical excision under general anesthesia. A subperiosteal dissection was performed in the posterior surface of the hyoid bone that was not transected and the mass was isolated just below the hyoepiglottic ligament and without entering the pharynx. The patient was discharged without any pain or complication on the first postoperative day.

**Figure 4 diagnostics-13-00932-f004:**
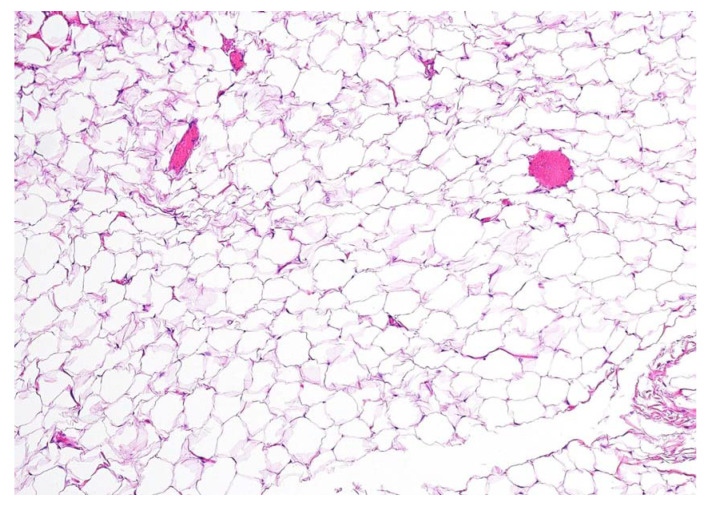
Definitive histopathological examination of the specimen confirmed the hypothesis of an encapsulated lipoma composed of simple adipose cells with no atypia present. In the present picture a 10× microscopic view of a formalin-fixed paraffin-embedded section of the lesion is shown, stained with hematoxylin-eosin. No thyroid remnants were apparent in all of the examined sections. No swallowing alterations were reported at a follow-up of 6 months after the operation, nor was there any clinical and ultrasonographic evidence of disease recurrence.

**Figure 5 diagnostics-13-00932-f005:**
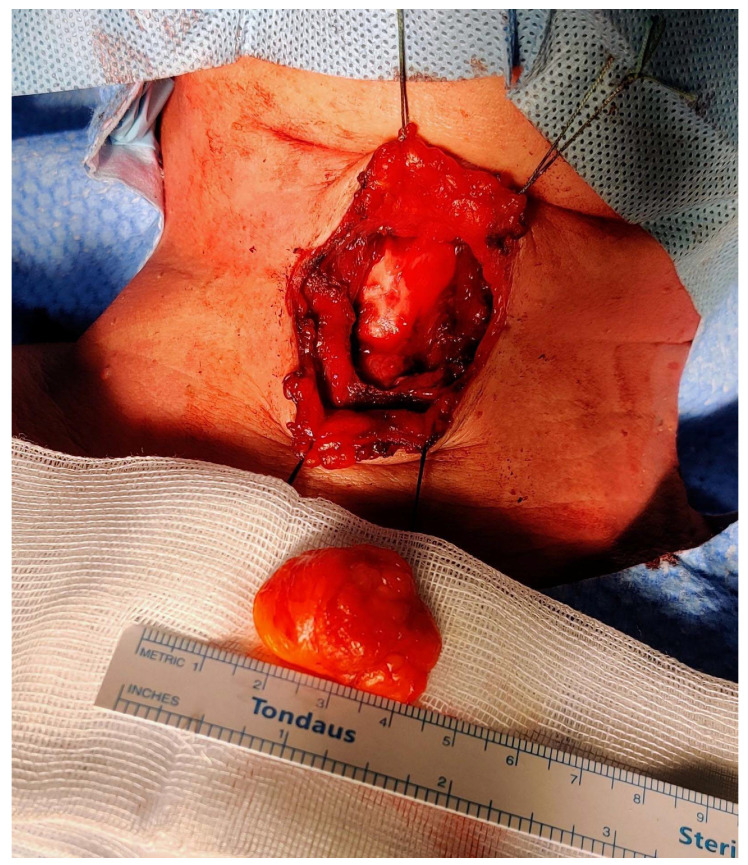
Intraoperative picture after the completion of the resection. The PubMed database was used in order to perform the review of the literature from the beginning to November 2022. The CARE (CAse REports) guidelines and its accompanying check-list served as a reference in the preparation of the present report [9]. The following search strings were used: “thyroglossal duct AND lipoma” (eight results); “thyroglossal AND lipoma” (18 results); “neck midline AND lipoma” (15 results). We retrieved a total of 41 articles. After the removal of duplicates and non-pertinent articles, only one paper was included [10]. Tsai and colleagues reported a single case of TGD lipoma, thus making our case the second one to be published in the available literature to the best of our knowledge. The authors described a 3.5 × 3.0 × 3.0 cm^3^ oval-shaped mass located just between the hyoid bone and the thyroid cartilage; despite a radiological appearance suspicious for lipoma (isoechoic mass at the US, density in Hounsfield units on the computed tomography scan compatible with fat), the authors performed a full Sistrunk operation and they declare no recurrences at 18 months of follow-up. Histopathological examination revealed a small rim of thyroid tissue adjacent to the mature adipocytes, while data on postoperative swallowing functions were not reported [10]. Instead, Sharudin and coworkers illustrated the case of a 41-year-old man who presented with a submucosal lipoma of the epiglottic valleculae [11]. Despite being potentially akin to our case, we decided not to consider it associated with the TGD tract because of its more posterosuperior localization and its probable origin from the fat occupying the pre-epiglottic space. Furthermore, that lesion presented as an oropharyngeal exophytic mass behind the foramen cecum, and it was amenable to transoral endoscopic resection by electrocautery [11]. Finally, Costa et al. published a case of a lipoma of the pre-epiglottic space causing chronic dysphagia in 2017: in this paper, however, the lesion was completely endolaryngeal (pre-epiglottic and left paraglottic space), no neck masses were apparent, and again transoral laser-assisted resection was sufficient to cure the patient [12].

**Figure 6 diagnostics-13-00932-f006:**
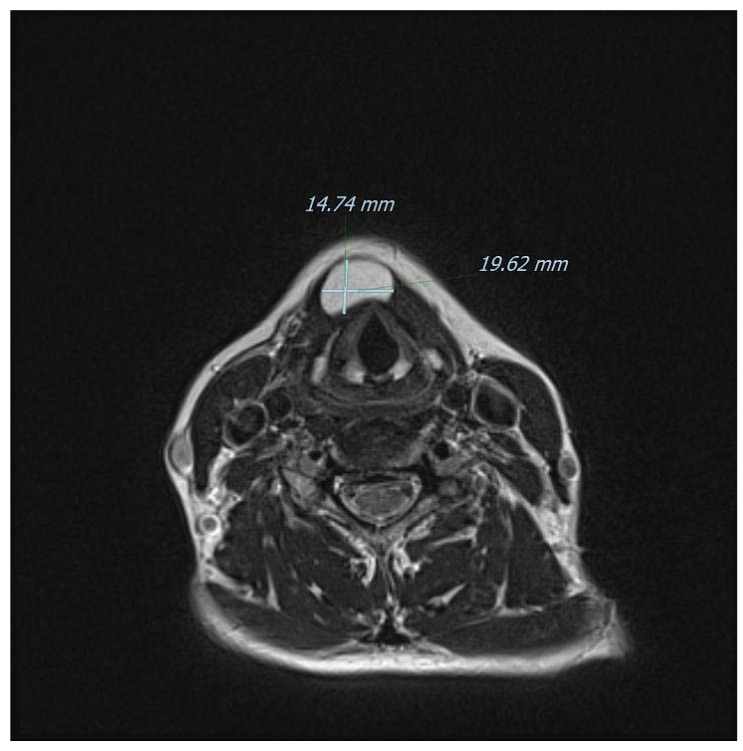
Another T2-weighted axial view of the mass on MRI. Ultrasonography is considered to yield a high diagnostic accuracy for subcutaneous lipomas (sensitivity of 88.1%; specificity of 99.3%) [13], but in our case, it was inaccurate in defining the boundaries of the lesion and therefore the surgical planning. Modern CT and MRI techniques offer instead a superb definition of the anatomical relationships as well as a very strong correlation with final histopathology, especially in children [14]. Furthermore, and for midline neck masses only, the role of fine-needle aspiration cytology/biopsy (FNAC/FNAB) remains debatable even for the common TGD cysts: for instance, a retrospective evaluation at the Johns Hopkins Department of Pathology found a diagnostic sensitivity of only 62%, with a calculated positive predictive value of 69% [15,16]. Additionally, imaging remains fundamental to exclude a rarely coexisting carcinoma (usually papillary-type), that usually occurs in patients 40 years of age or older. In detail, classically described features of a suspected malignancy include enhancing solid nodular masses (more frequently observed) and/or calcifications (more specific) within the TGD cyst [6]. Sistrunk’s operation remains the mainstay of management for TGD cysts because it has been shown to reduce the risk of recurrence down to 3–10% of adult cases, [17,18,19] and to around 20% in children [20]. There is also a non-negligible rate of postoperative complications and, in a recent large pediatric American series, postoperative rates of cervical seroma, local infection, and need for surgical revision were reported to be 9.5%, 7%, and 7.6%, respectively [21]. For other conditions associated with the TGD remnants instead, the evidence regarding the best surgical approach is weaker. For example, midline dermoid cysts usually require simple excision without sacrificing the hyoid bone and a preoperative differential diagnosis is now feasible by exploiting the diffusion-weighted imaging (DWI) technique with MRI, or even with ultrasound alone [22,23]. Interestingly, the two lesions may in very rare cases occur at the same time; therefore, some surgeons prefer to perform Sistrunk’s operation for every central (pediatric) neck mass [24,25]. The absence of thyroid tissue in our case might be viewed as a complete lipomatous degeneration of a TGD remnant, as speculated by Tsai and colleagues [10]. However, we currently have no preoperative imaging methods to determine whether thyroid cells are present in these kinds of neck masses unless they are metabolically active and therefore visible on scintigraphy [26]. In the present case, scintigraphy was not performed nor it was available in the study by Tsai et al. [10]. Indeed, the use of a “blind scintigraphy” may constitute an interesting approach in the work-up of midline masses, but the available experience in this regard is limited. It must also be remembered that even in classical TGD cysts, thyroid tissue in the form of follicles or scattered cells are found only in around 50% of cases [27].

## Data Availability

Data is available upon a reasonable request from the corresponding author.

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
