# Peer review of "Thyroglossal Duct Lipoma: A Case Report and a Systematic Review of the Literature for Its Management"

_diagnostics, 2023, doi:10.3390/diagnostics13050932_

Round 1
Reviewer 1 Report
1. Different parts of the article have not been determined. It seems better to separate part of your valuable article as the introduction,...
2. Reference number 9 has not been inserted appropriately.
3. The diagnosis of this case was based on the pathology report. Please insert figures of pathology.
4. Despite this case being challenging to manage, as mentioned in the title, not enough data is shown in your review. Please discuss it further.
5. In the figure caption, please interpret them. It needs to rewrite again.
6. Please make sure that there are not any grammar or punctuation mistakes.
Reviewer 2 Report
Thank you for this opportunity to review this case report; In my opinion, this is a good case report and can add valuable data to the readers of this journal.
Author Response
Dear Reviewer, thank you for your kind comments on our work.
Reviewer 3 Report
Since you mention the US aspect, please do enclose both US aspect and also scintigrahy. Since the US and Ct were highly suggestive for a TGD pathology, scintigraphy is one of the first line imagistic evaluations. The mentioned ”blind” scintigraphy aspect is worth to be shown in the manuscriot.
Author Response
Dear Reviewer, thanks for the useful comments on our manuscript. Unfortunately, the US had been performed by a private practice radiologist and we were not able to retrieve anything but the written report. Scintigraphy instead was not performed in the present case because it is not standard practice to perform it even in the frank TGD cysts. However we did address this point at the end of the revised manuscript.
Round 2
Reviewer 1 Report
1. Different parts of the article have not been determined. It seems better to separate part of your valuable article as the introduction,...
2. The diagnosis of this case was based on the pathology report. Please insert figures of pathology.
3. Despite this case being challenging to manage, as mentioned in the title, not enough data is shown in your review. Please discuss it further.
4.In the figure caption, please interpret them as some words. It needs to rewrite again.
Author Response
Response to Reviewer 1 Comments
- Different parts of the article have not been determined. It seems better to separate part of your valuable article as the introduction,...
Dear Reviewer, again sorry for the misunderstanding but although you are right that no clear distinction in terms of Introduction, MM, Results and Conclusions is present in the manuscript, this is not possible in this case. However, these are the requirements of the “Interesting Images” format, which I copy here for more clarity: “Interesting Images: Diagnostics encourages the submission of Interesting Images. The number of images are at the discretion of the author. No regular manuscript text (introduction/methods/results/discussion) should be included. Instead, images should be accompanied by detailed legends with no restriction in length. Reference citations should appear in the legends. Also, an unstructured abstract of no more than 200 words should be included as well as a list of 3 to 10 keywords. Image files can be included either in the template or uploaded separately in high resolution. There are no restrictions on use of colour or image size, however features should be sharp and not blurred. For readability, we recommend that any text in figures is at least 12 pt in size. Submitted images will be peer-reviewed under the same process as a regular research article.”
- The diagnosis of this case was based on the pathology report. Please insert figures of pathology.
Dear Reviewer, as previously mentioned, the pathological picture of the lipoma is microscopically of little significance because the very “interesting images” (as requested by the format) are those from the radiologists. In addition, we are currently experiencing some troubles with our pathologists and we need to wait some weeks to let them retrieve this picture from the digital archive.
- Despite this case being challenging to manage, as mentioned in the title, not enough data is shown in your review. Please discuss it further.
In the initial draft of the manuscript, we had planned to create a PRISMA flowchart, so as to present all the data extracted from the literature search in a table. Eventually, because of the format of the present submission and because of the paucity of the extracted data (the only real other thyroglossal lipoma was the one from Tsai et al.), we decided to describe our findings in a descriptive manner, which we believe is well coordinated with all the pictures included.
4.In the figure caption, please interpret them as some words. It needs to rewrite again.
Dear Reviewer, we are really and deeply sorry but we still do not understand this comment. Which figure caption are we supposed to rewrite more clearly? In addition, because of the very format of the article there are no actual figure captions which are the text itself.

Round 3
Reviewer 1 Report
Thank you for your recent response. I appreciate you taking the time to write such a helpful letter. Wishing you the best in the following scientific phase of your life.